# Ingestion of artificial sweeteners leads to caloric frustration memory in *Drosophila*

Pierre-Yves Musso[1,3], Aurélie Lampin-Saint-Amaux[1], Paul Tchenio[1,2] & Thomas Preat [1]

Non-caloric artificial sweeteners (NAS) are widely used in modern human food, raising the question about their health impact. Here we have asked whether NAS consumption is a neutral experience at neural and behavioral level, or if NAS can be interpreted and remembered as negative experience. We used behavioral and imaging approaches to demonstrate that *Drosophila melanogaster* learn the non-caloric property of NAS through post-ingestion process. These results show that sweet taste is predictive of an energy value, and its absence leads to the formation of what we call Caloric Frustration Memory (CFM) that devalues the NAS or its caloric enantiomer. CFM formation involves activity of the associative memory brain structure, the mushroom bodies (MBs). In vivo calcium imaging of MB-input dopaminergic neurons that respond to sugar showed a reduced response to NAS after CFM formation. Altogether, these findings demonstrate that NAS are a negative experience for the brain.

[1] Genes and Dynamics of Memory Systems, Brain Plasticity Unit, CNRS, ESPCI Paris, PSL Research University, 75005 Paris, France. [2] Laboratoire Aimé Cotton, CNRS, Université Paris-Sud, 91400 Orsay, France. [3] Present address: Department of Zoology, Cell and Developmental Biology, University of British Columbia, Vancouver, BC, Canada V6T 1Z3. Paul Tchenio and Thomas Preat jointly supervised this work. Paul Tchenio Deceased Correspondence and requests for materials should be addressed to T.P. (email: thomas.preat@espci.fr)

Obesity and diabetes rates have been increasing for several decades in the human population, and an excessive sugar diet is well-known to promote metabolic disorders. For this reason, the food-processing industry introduced almost one century ago the use of non-caloric artificial sweeteners (NAS) into food[1]. These NAS can sweeten food without the caloric content usually present in sugars. From a physiological point of view, most NAS transit through the digestive tract without being modified, and thus they do not provide any source of energy[2,3]. Here we have used *Drosophila melanogaster*, a suitable model organism for feeding behavior and nutrient sensing studies[4–6], to investigate the effect of NAS on physiology.

Sugars are widely used as an unconditioned positive stimulus in associative memory experiments, in which they can trigger a response in starved animals such as the proboscis extension response in insects[6–9] or feeding behavior in mammals[10]. This unconditioned stimulus can be associated to a conditioned stimulus that does not alone trigger a reflex response in animals, such as odorants. After pairing, the subsequent presentation of the conditioned stimulus will then elicit a response similar to that of the unconditioned stimulus alone. Thus, an odor previously paired to a sugar will elicit an approach behavior. When an odorant is associated with a sugar in *Drosophila*, two types of appetitive memory can be formed that rely on different sugar properties: sugar palatability, which leads to the formation of short-term memory (STM), and sugar caloric content, which is an essential component for the formation of long-term memory (LTM)[11–13]. More specifically, when the non-caloric sugar L-glucose (an enantiomer of the natural D-glucose with a similar taste[7,14] but which cannot be metabolized[15–17]) is paired with an odor, appetitive STM forms but not LTM (Supplementary Fig. 1a, b). In spite of this, we previously showed that flies will form LTM after L-glucose conditioning if an energy source is provided up to 5 h after experiencing a sweet taste[13]. Appetitive STM and LTM both form after a single cycle of training, although LTM depends specifically on de novo protein synthesis[18,19], an energy-costly process[20]. Interestingly, the energy income that follows digestion of caloric sugars is also a requirement for LTM formation in mammals[10,21]. These general observations on appetitive learning raise a number of important questions concerning NAS. For instance, the brain expects an energy income after sensing a sweet taste, but what happens when there is no income? Moreover, can animals learn and remember that these sweet-tasting sugars are not nutritious? And if so, how long does this memory last and what are the underlying mechanisms?

Here we examined whether NAS consumption may comprise an unnatural feeding condition that is remembered as a negative experience. As a general approach, we first fed flies with non-nutritious L-glucose. Then, to estimate the potential memory of a negative experience, we performed classical olfactory appetitive conditioning consisting of pairing L-glucose with an odorant. Our results demonstrate that exposing flies to L-glucose leads to a loss in the positive value of this sugar. This devaluation constitutes a long-lasting memory that depends on the functional connectivity between the mushroom bodies (MBs), i.e., the major learning and memory structure in *Drosophila*, and the dorsal paired medial (DPM) neurons. Finally, our calcium imaging experiments reveal that the dopaminergic neurons, which convey positive cue information about sugar to the MBs, exhibit a decreased response to L-glucose or nutritious D-glucose after ingesting the non-nutritious sugar. Our results establish that flies learn to differentiate when a sweet taste experience is not followed by a nutritious income, and that the formed memory leads to a diminished sugar value.

## Results

**Flies learn to devalue NAS after experiencing them.** We speculated if flies could learn the non-nutritious nature of NAS. If this were the case, one could expect that after experiencing a non-nutritious sugar it would be devalued. To determine the extent of any devaluation, we exposed starved flies to L-glucose for 1 min. Subsequently, flies were trained 1 day later with an odorant associated with L-glucose, and their olfactory STM was assayed 2 h after training. In this protocol, the olfactory memory test serves as an L-glucose devaluation test. We anticipated that flies deceived by non-nutritious L-glucose would display lower olfactory memory scores. Indeed, flies pre-fed for 1 min with L-glucose displayed significantly lower olfactory STM scores (Fig. 1a), suggesting that flies can devalue L-glucose after first experiencing it. To further assess whether the decreased score was due to the non-caloric property of the pre-fed sugar, we performed control experiments in which flies were pre-fed with either regular medium or natural D-glucose. In either case, we observed high STM scores equivalent to that of the non-pre-fed flies (Fig. 1a), suggesting that L-glucose specifically triggers the devaluation. Pre-feeding flies with L-glucose did not affect their response to L-glucose in the absence of training (Fig. 1b and Supplementary Fig. 1c) or olfactory acuity (Supplementary Fig. 1d). Thus, the low olfactory memory score is neither due to impaired sugar perception nor to impaired olfaction. To further characterize the devaluation process, we tested immediate olfactory memory after pre-feeding. Indeed, a defect was observed immediately after olfactory conditioning (Supplementary Fig. 1e). Altogether, these results indicate that flies learn to devalue L-glucose due to its non-nutritious nature.

A key aspect of the devaluation effect to determine is whether it involves early sensory (gustatory) information, or rather a brain function following sugar ingestion. The observation that flies pre-fed with L-glucose display a normal reaction to L-glucose in proboscis extension experiments (Fig. 1b) and sugar attraction tests (Supplementary Fig. 1c) suggests that devaluation is not a sensory process. We thus hypothesized that the devaluation after L-glucose pre-feeding was linked to a lack in post-ingestion processing of nutritional food content. To demonstrate this we utilized phlorizin, a specific antagonist of the intestinal glucose transporter[16] that has been successfully used to block glucose entry into hemolymph in *Drosophila* and to prevent LTM formation after D-glucose conditioning[13]. Interestingly, pre-feeding flies with a mixture of D-glucose and phlorizin reproduced the devaluation effect of L-glucose pre-feeding (Fig. 1a), whereas both the L-glucose response and olfaction remained normal (Fig. 1b and Supplementary Fig. 1c, d). To further demonstrate that devaluation is triggered by a lack of post-ingestion processing of nutritious content, we hypothesized that adding back nutritious food within a few minutes after L-glucose pre-feeding should impair devaluation. Indeed, re-feeding flies with either D-glucose or classical medium after L-glucose pre-

**Fig. 1** Flies learn to devalue sugar value through taste recognition and a post-ingestion mechanism. **a** Pre-feeding flies for 1 min with non-nutritious L-glucose 24 h before olfactory conditioning with the same sugar induces significantly lower STM scores as compared to non-pre-fed flies or flies pre-fed with classical medium ($F_{(2,52)} = 12.63$; $p < 0.0001$; $n \geq 18$; $p > 0.999$ in post hoc analysis between non-pre-fed flies and flies pre-fed with medium). In contrast, pre-feeding flies with nutritious D-glucose does not affect olfactory memory scores as compared to non-pre-fed flies or flies pre-fed with medium ($F_{(2,52)} = 0.553$; $p = 0.578$; $n \geq 18$). However, pre-feeding flies with a mixture of D-glucose and phlorizin induces significantly lower olfactory memory scores as compared to D-glucose pre-fed flies (t-test, $t_{37} = 2.544$; $p = 0.015$; $n \geq 19$). **b** PER (Proboscis Extension Response) scores with increasing concentrations of L-glucose in flies pre-fed with L-glucose or a mixture of D-glucose + phlorizin 24 h are similar to flies pre-fed with D-glucose or medium, as well as non-pre-fed flies ($F_{(16,1032)} = 0.906$; $p = 0.494$; $n \geq 47$). **c** Flies pre-fed and conditioned with arabinose display significantly lower memory scores as compared to non-pre-fed flies conditioned with arabinose (t-test, $t_{42} = 2.056$; $p = 0.046$; $n \geq 21$). **d** Pre-feeding flies with L-glucose and conditioning them with arabinose induces significantly higher memory scores in comparison to flies pre-fed and conditioned with L-glucose (t-test, $t_{18} = 2.188$; $p = 0.042$; $n = 10$). See also Supplementary Fig. 1. Means are ± SEM; statistical tests: t-test, one-way ANOVA, two-way ANOVA, and RM-ANOVA; NS: $p \geq 0.05$; *$p < 0.05$ in comparison between two groups for t-test and in post hoc comparisons with other groups for ANOVA

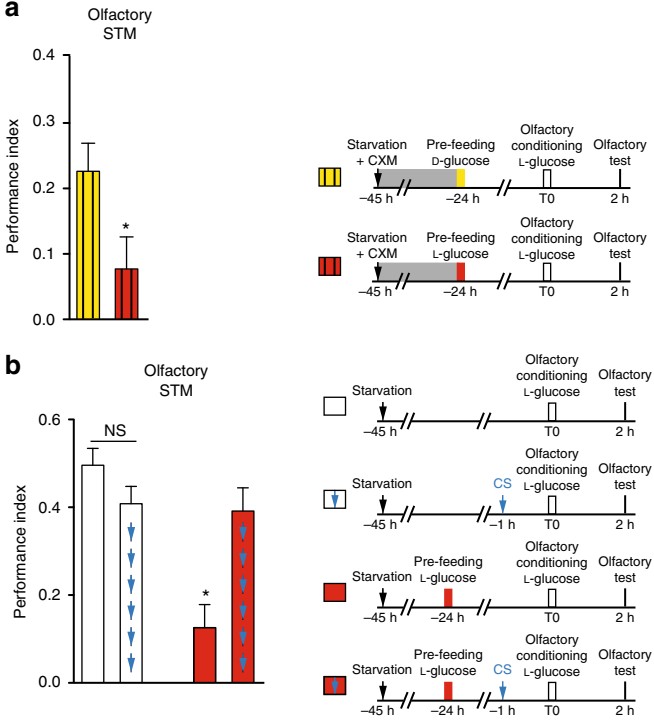

**Fig. 2** CFM processing does not require de novo protein synthesis and is sensitive to anesthesia treatment. **a** Treating flies with the protein synthesis inhibitor CXM before pre-feeding does not alter the negative effect of L-glucose pre-feeding (t-test, $t_{37} = 2.301$; $p = 0.027$; $n \geq 19$). **b** Cold-shock application 1 h prior to olfactory conditioning did not impair the 2-h olfactory memory scores (t-test, $t_{22} = 1.592$; $p = 0.125$; $n = 12$). Flies pre-fed with L-glucose and conditioned with L-glucose display lower olfactory memory scores as compared to non-pre-fed flies (t-test, $t_{22} = 5.687$; $p < 0.001$; $n = 12$). Applying cold shock treatment to flies pre-fed and conditioned with L-glucose abolished CFM (t-test, $t_{22} = 3.572$; $p = 0.001$; $n = 12$). See also Supplementary Fig. 2. Means are ± SEM; statistical test: t-test; NS: $p \geq 0.05$; *$p < 0.05$ in comparison between two groups

feeding prevented devaluation (Supplementary Fig. 1f). Furthermore, mixing increasing concentrations of the tasteless but nutritious D-sorbitol with L-glucose during pre-feeding prevented devaluation in proportion to its concentration (Supplementary Fig. 1g).

Together, these results demonstrate that pre-exposure to a sugar that cannot be metabolized leads to devaluation of its associative sugar value. This memory lasts for at least 24 h and it ensures that olfactory appetitive learning does not occur with a previously encountered non-nutritious sugar.

To determine whether this devaluation is specific to L-glucose or if it could be a more general property of non-nutritious sugars, we repeated the pre-feeding protocol using another non-nutritious sugar, arabinose[11,12,22]. This yielded the same devaluation phenomenon (Fig. 1c), indicating that it is not specific to L-glucose. Since the devaluation memory forms when the experience of a sweet taste is not followed by the expected caloric income, we have referred to it as Caloric Frustration Memory (CFM).

Different sugars elicit different calcium responses in neurons involved in the sweet signaling pathway[23–28], in addition to different proboscis extension reflexes[7,8,29]. These findings indicate that flies are able to discriminate between different sugars. Accordingly, one could expect that flies will show sugar devaluation only when the same NAS is used for pre-feeding and olfactory conditioning. To address this, we pre-fed two

groups of flies on L-glucose. For the first group, flies were conditioned with L-glucose as a positive control, whereas the second group was trained with arabinose. As expected, flies pre-fed and conditioned on L-glucose displayed low STM, whereas flies pre-fed on L-glucose and conditioned on arabinose displayed normal olfactory memory (Fig. 1d). This result demonstrates that CFM is specific to the NAS taste and that flies do not generalize the devaluation.

**CFM is a new form of long-lasting memory**. Our protocol had a 24-h delay between pre-feeding and olfactory conditioning, leaving open the possibility that CFM could be a consolidated memory that involves de novo protein synthesis, as occurs with appetitive olfactory LTM. To address this, we next assessed the sensitivity of CFM to treatment by cycloheximide (CXM), a protein synthesis inhibitor widely used to characterize olfactory LTM in Drosophila[18,19,30]. If CFM is protein synthesis dependent, one would expect an impaired devaluation, i.e., the olfactory memory score of groups treated with CXM and pre-fed on either D- or L-glucose would be similarly high. Surprisingly, the scores of L-glucose pre-fed flies remained significantly lower than those of D-glucose-trained flies (Fig. 2a), suggesting that long-lasting CFM does not depend on de novo protein synthesis. To verify the action of CXM, a control experiment was performed in parallel in which flies were trained for classical appetitive LTM and pre-treated with CXM. As expected, the CXM-treated group displayed a significantly lower memory score (Supplementary Fig. 2), demonstrating the efficiency of the CXM treatment. These results indicate that CFM is a long-lasting memory that does not depend on de novo protein synthesis. To further characterize CFM, we examined whether it is resistant to an anesthesia treatment such as cold shock[18,30,31]. We first checked that a 4 °C cold shock applied 1 h before olfactory conditioning did not affect the 2-h olfactory memory in flies that were not pre-fed with L-glucose (Fig. 2b). We then applied a cold shock treatment prior to the olfactory conditioning of flies pre-fed with L-glucose. Interestingly, anesthesia abolished CFM, as these flies displayed a significantly higher score than non-anesthetized L-glucose pre-fed flies (Fig. 2b). CFM is therefore sensitive to anesthesia, indicating that it does not correspond to either LTM, which is resistant to cold shock[18], or to the so called Anesthesia-Resistant Memory[18,30,31], but rather to a unique form of long-lasting memory that probably relies on the activity of recurrent neuronal circuits.

**CFM formation requires the activity of DPM, PAM and MBs**. Next, we focused on which brain circuits are involved in CFM processing. Associative olfactory memories are encoded in the 4000 Kenyon cells (KC) of the MBs. The KCs project their axons vertically and horizontally, forming the vertical and medial lobes, respectively. Projection neurons convey sensory olfactory information from the antennal lobes to the MB calyx[32], i.e., the KC dendrites. In turn, PAM dopaminergic neurons convey sweet sensory information to the MB lobes[12,22,33,34]. Since CFM results from the association between sweet taste and the lack of energy income, we considered that it might involve the MBs. We first investigated if CFM requires MBs by inhibiting their synaptic activity during pre-feeding as well as 2.5 h after pre-feeding, a period potentially corresponding to the association between the sweet taste stimulus and the absence of an energy income. Expression of the dominant negative thermo-sensitive shibire protein (Shi[ts]) in a given set of neurons allows the blockade of their neurotransmission at the restrictive temperature (33 °C), which is then released at the permissive temperature (25 °C). By expressing Shi[ts] in MB neurons using the highly MB-specific VT30559-GAL4 driver[35], we observed that CFM was impaired

when MB output was blocked during this 3-h window (Fig. 3a). Subsequently, we checked that CFM was normal in flies pre-fed at the permissive temperature (Supplementary Fig. 3a), and that the response to L-glucose was normal at the restrictive temperature (Supplementary Fig. 3b). These results demonstrate that MB neuronal output is required during pre-feeding and/or up to 2.5 h later. To further investigate the role of MBs in CFM formation, we used GAL4 drivers targeting the three main neural populations of MBs: γ neurons (using *VT049483-GAL4*); α/β neurons (using *MB008B-GAL4*); and α'/β' neurons (using *VT030604-GAL4*). Synaptic transmission blockade of a single sub-population of MBs neurons did not lead to CFM impairment (Supplementary Fig. 3c–e), suggesting that at least two MB neuronal populations can sustain CFM formation.

To further investigate the nature of CFM and to confirm that it does not correspond to a classical form of LTM that relies on de novo protein synthesis, we inhibited the expression of the CREB transcription factor in adult MBs neurons with RNAi and the thermo-inducible TARGET system (which uses the *tub-GAL80^ts; VT30559-GAL4* driver)[35,36]. As for the CXM experiment, inhibition of CREB in adult MBs did not affect CFM (Supplementary Fig. 4a). A control experiment was then performed in parallel to verify the action of CREB on LTM inhibition, in which flies were trained for classical appetitive LTM. As expected, flies with CREB-inhibited MBs displayed impaired olfactory LTM (Supplementary Fig. 4b; see Supplementary Fig. 4c for control).

We then investigated the circuits involved in CFM formation. The DPM neurons consist of a pair of both serotoninergic and GABAergic neurons projecting to all MB lobes, with the particularity that they are both afferent and efferent to the MBs[37,38]. Activity in the DPM and MBs is reported to be involved in both long-lasting aversive and appetitive memory formation[37,39–42]. Since CFM is a long-lasting memory requiring MB synaptic activity, we wondered whether it shares other neural network similarities with already characterized long-lasting memories. We therefore investigated DPM involvement in CFM using Shi^ts and the *VT64246-GAL4* driver. DPM blockade 30 min before, during and 1.5 h after pre-feeding was sufficient to impair CFM (Fig. 3b). We then checked that CFM was normal in flies pre-fed at the permissive temperature (Supplementary Fig. 4d), and that the response to L-glucose was normal at the restrictive temperature (Supplementary Fig. 4e). These results demonstrate that the DPM neurons are required during pre-feeding and/or the subsequent period lasting up to 1.5 h. Altogether, our results suggest the existence of a functional loop between the MBs and DPM that is required for the formation of CFM.

The MB neurons receive input from neurons conveying sweet taste information[33], and in turn respond to sweet stimulation[43]. One previous study concluded from in vivo calcium imaging of DPM neurons that these neurons do not respond to sugar stimulation[42]. However, this claim is somewhat surprising since DPM are required for appetitive memories[18,39,42,44]. We reinvestigated the issue of the DPM response to sugar stimulation using the *GCaMP6f* calcium probe and the *VT64246-GAL4* DPM-specific driver[37,45,46]. Interestingly, this experiment revealed that DPM neurons respond significantly to non-nutritious L-glucose

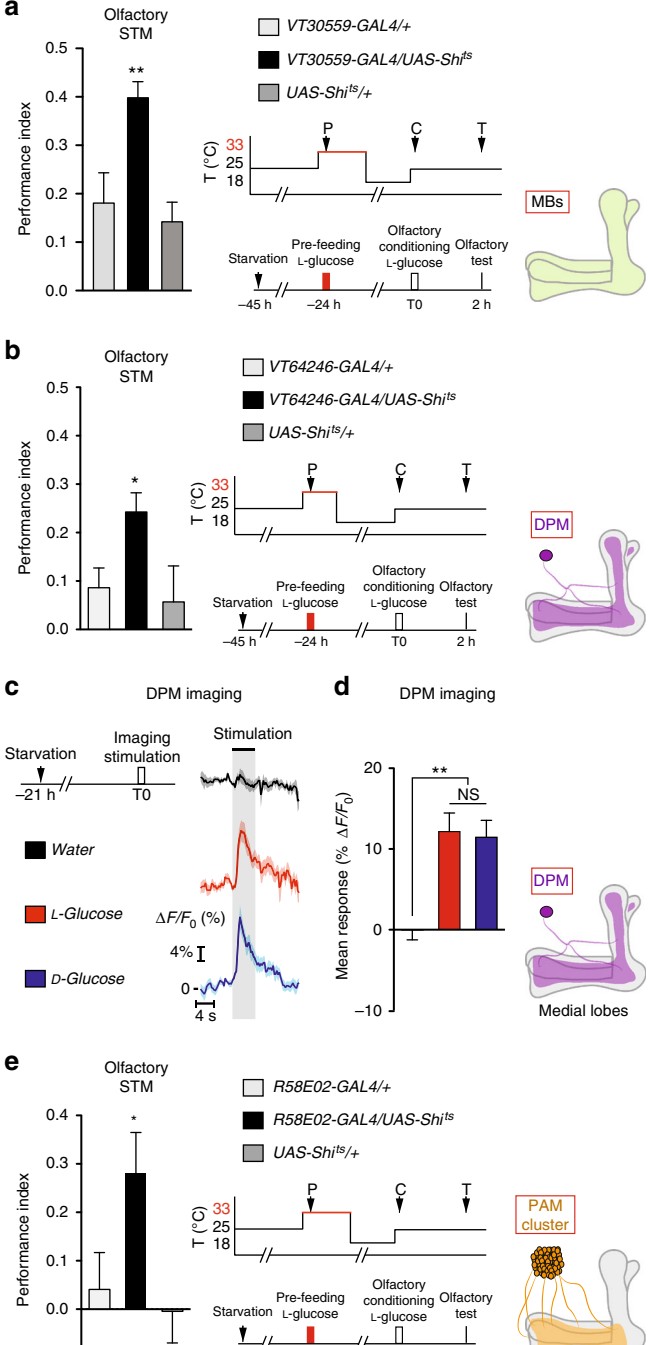

**Fig. 3** CFM formation requires synaptic activity in DPM neurons, the MB and PAM neurons. **a** Blocking MB neurons under *VT30559-GAL4* control 30 min before, during, and until 2.5 h after L-glucose pre-feeding abolishes the sugar devaluation ($F_{(2,42)} = 8.966$; $p < 0.001$; $n \geq 14$). **b** Blocking DPM neurons under *VT64246-GAL4* control 30 min before, during and until 1.5 h after L-glucose pre-feeding abolishes the sugar devaluation ($F_{(2,44)} = 3.868$; $p = 0.028$; $n \geq 14$). **c**, **d** DPM neurons respond selectively to sweet stimulation. **c** Left: imaging protocols; right: time course of response. $n \geq 14$. Black bar: stimulus presentation. **d** Average response to L-glucose, D-glucose and water. DPM neurons responded significantly to L- and D-glucose ($t$-test, $t_{12} = 5.290$; $p < 0.001$; for L-glucose and $t$-test, $t_{18} = 5.450$; $p < 0.001$ for D-glucose; $n \geq 134$) at significantly higher levels than water ($F_{(2,42)} = 9.090$; $p < 0.001$; $n \geq 11$; $p = 0.966$ in post hoc comparison between the responses to D- and L-glucose). **e** Blocking PAM neurons under *R58E02-GAL4* control 30 min before, during and until 1.5 h after L-glucose pre-feeding abolishes the sugar devaluation ($F_{(2,25)} = 4.066$; $p = 0.03$; $n \geq 9$). See also Supplementary Figs. 3, 4. Means are ± SEM; statistical tests: $t$-test and one-way ANOVA; NS: $p \geq 0.05$; *$p < 0.05$; **$p < 0.01$ in comparison between two groups for $t$-test and in post hoc comparisons with both parental controls for ANOVA

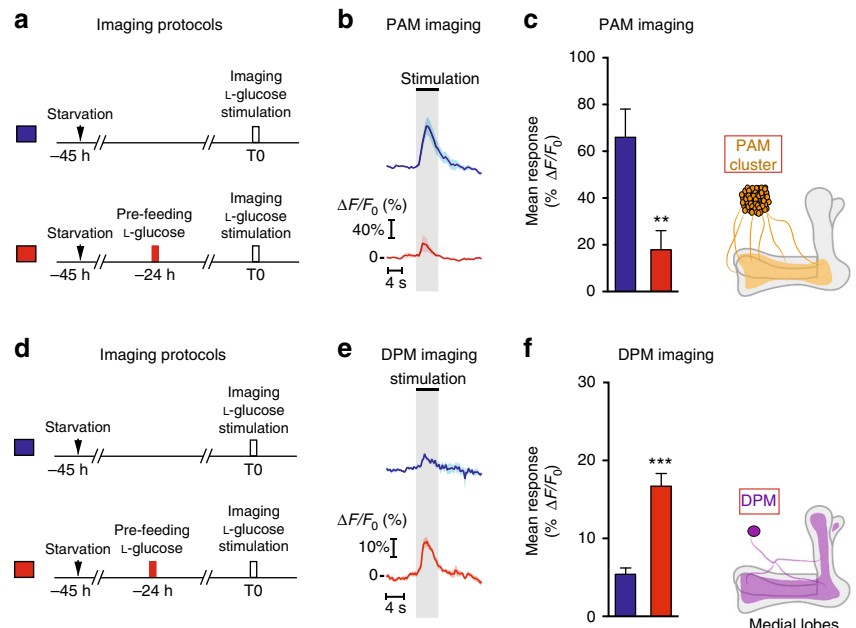

**Fig. 4** PAM dopaminergic neurons display lower calcium responses while DPM neurons display a higher calcium response to NAS after pre-feeding. **a** Pre-feeding protocol used before the imaging experiment: flies were pre-fed or not with L-glucose for 1 min, 24 h before L-glucose stimulation under the microscope. **b** Time course response of PAM neurons. $n = 10$. Black bar: stimulus presentation. **c** Average response to L-glucose. Flies pre-fed with L-glucose displayed a significantly lower response in comparison to non-pre-fed flies ($t$-test, $t_{18} = 3.03$; $p = 0.004$; $n = 10$). **d** Pre-feeding protocol used before the imaging experiment: flies were pre-fed or not with L-glucose for 1 min, 24 h before L-glucose stimulation under the microscope. **e** Time course response of DPM neurons projecting on MBs medial lobes. $n = 8$. Black bar: stimulus presentation. **f** Average response to L-glucose. Flies pre-fed with L-glucose displayed a significantly higher response in comparison to non-pre-fed flies ($t$-test, $t_{14} = 6.2$; $p < 0.001$; $n = 8$). See also Supplementary Fig. 5. Means are ± SEM; statistical test: $t$-test; NS: $p \geq 0.05$; **$p < 0.01$; **$p < 0.001$ in comparison between two groups

and nutritious D-glucose, in starved flies (Fig. 3c, d). The two sugars elicited similar response levels, whereas the DPM neurons did not respond to water (Fig. 3c, d), showing that the increased activity is due to sugar stimulation. This discrepancy with the previous study[42] might be due to the fact that the authors used a less sensitive calcium reporter, *GCaMP1.6*, which is known to yield lower amplitude responses than *GCaMP6f*[47−49].

During appetitive learning, when flies consume the sugar associated with an odor, PAM dopaminergic neurons afferent to the MBs convey sweet information to the MBs[12,33,34,50]. We hypothesized that this information is necessary to the MBs to encode the sweet identity from the food source, and hence should be required during CFM formation. Indeed, PAM blockade with Shi[ts] 30 min before, during and 1.5 h after pre-feeding was sufficient to impair CFM (Fig. 3e). We checked that CFM was normal in flies pre-fed at the permissive temperature (Supplementary Fig. 4f), and that the response to L-glucose was normal at the restrictive temperature (Supplementary Fig. 4g). These results demonstrate that PAM neurons are required during pre-feeding and/or the subsequent period lasting up to 1.5 h. Altogether, our results suggest the existence of a functional loop between the MBs, the DPM and PAM that is required for the formation of CFM.

**CFM inhibits PAM neuronal response to sugar presentation**. Following the pre-feeding step with L-glucose, flies learn that the ingested sugar is not nutritious. The brain then retrieves this information during the olfactory conditioning that occurs 24 h later, preventing the formation of a strong appetitive olfactory memory with the non-nutritious sugar. A key issue is whether CFM inhibits olfactory memory at an early step, thus preventing its formation upstream of the MBs, or if CFM antagonizes olfactory memory in the MBs. During appetitive learning, when

flies consume the sugar associated with an odor, PAM dopaminergic neurons afferent to the MBs convey sweet information to the MBs[12,33,34,50]. We investigated if the devaluation induced by L-glucose pre-feeding was detectable at the level of the PAM neuronal response to L-glucose after pre-feeding. We therefore imaged the terminals of PAM neurons on MB lobes in vivo by expressing a calcium reporter in PAM neurons with the *58E02-GAL4* driver[12,33]. For the training protocol, one group of flies was pre-fed on L-glucose while the other was not. 24 h later, the response of PAM neurons was imaged during L-glucose stimulation (Fig. 4a). Non-pre-fed flies displayed a high positive calcium response, whereas flies pre-fed on L-glucose displayed a significantly lower response (Fig. 4b, c). To reinforce this intriguing result, we imaged PAM neurons after various pre-feeding conditions (Supplementary Fig. 5a). The PAM response to L-glucose was normal after pre-feeding on either D-glucose or medium (Supplementary Fig. 5a–c), two conditions that do not lead to CFM (see Fig. 1). On the contrary, flies pre-fed with a mixture of D-glucose and phlorizin, which does lead to CFM, displayed a decreased response (Supplementary Fig. 5b, c). These results indicate that the sugar devaluation occurs directly at the level of the PAM neurons. To confirm the taste specificity of CFM, we examined the PAM response to arabinose after L-glucose pre-feeding (Supplementary Fig. 5d), which did not lead to CFM (see Fig. 1c). L-glucose pre-fed flies and control (non-pre-fed) flies displayed equivalent responses to arabinose (Supplementary Fig. 5e, f). Providing energy after L-glucose pre-feeding prevented CFM formation (Supplementary Fig. 1f). To reinforce this result, we imaged PAM neurons after pre-feeding flies with L-glucose followed by D-glucose feeding (Supplementary Fig. 5g). As expected, PAM response was significantly higher in flies pre-fed on L-glucose and re-fed on D-glucose than in flies pre-fed on L-glucose (Supplementary Fig. 5h, i). Altogether, these results

confirm that CFM forms after non-caloric sugar pre-feeding and affects PAM response to sugar.

The role of DPM during CFM retrieval was further addressed by investigating if the devaluation induced by L-glucose pre-feeding was detectable at the level of the DPM response to L-glucose. For this, one group of flies was pre-fed on L-glucose while the other was not. 24 h later, the response of DPM neurons was imaged during L-glucose stimulation (Fig. 4d). Non-pre-fed flies displayed a low positive calcium response, whereas flies pre-fed on L-glucose displayed a significantly higher response (Fig. 4e, f). Altogether, these results suggest that CFM leads to an increased response in DPM neurons that may inhibit the response of PAM neurons to the devalued sugar, before the sugar information is processed in the MBs and associated to the olfactory modality.

**CFM affects the value of a nutritious sugar.** Exposure to NAS leads to CFM, which is revealed if the non-nutritious sugar encountered during olfactory learning displays the same taste. We therefore hypothesized that CFM could affect the value of a nutritious sugar during olfactory conditioning if this sugar has a similar taste to that of the NAS used for pre-feeding. Flies were pre-fed with L-glucose 1 day before an olfactory conditioning with D-glucose, and their immediate memory was assessed. Strikingly, flies pre-fed on L-glucose and conditioned on D-glucose displayed low olfactory memory scores, similar to that of L-glucose pre-fed flies conditioned on L-glucose (Fig. 5a). As expected, non-pre-fed control flies that were conditioned on L- and D-glucose displayed high olfactory memory scores, as did flies pre-fed with D-glucose (Fig. 5a). Remarkably, pre-feeding on L-glucose induced a significantly lower PAM response to D-glucose stimulation, 24 h later (Fig. 5b–d). These results demonstrate that CFM can devalue a sugar even though it is nutritious, indicating that NAS can therefore have a wide effect on brain physiology.

## Discussion
We have used a novel pre-feeding protocol to demonstrate the existence in the *Drosophila* brain of a mechanism that devalues food value after exposure to NAS. Sugars have two distinct properties involved in associative olfactory learning: the sweet taste that allows STM formation, and the energy that permits LTM formation[11,13,22,26,33]. Our previous studies established that once a sweet taste has been associated to an odorant, the energy can be delivered up to 5 h later in order to allow LTM formation. This dissociation strongly suggests that sweet taste and energy are two independent stimuli for appetitive LTM formation. Importantly, the present work provides new insight into the relationship between energy metabolism and memory, as we show that the fly brain is also able to encode and remember (for at least 24 h) the lack of energy associated with an NAS. Following pre-feeding with L-glucose or arabinose for only one minute, flies are able to learn that the sweet gustatory stimulus of this particular molecule is not associated with an energy income, and a CFM is thus formed. During subsequent presentation of the NAS, CFM is retrieved and the NAS appears devalued. Thus, when the NAS is presented in association with an odorant 24 h after pre-feeding, the olfactory learning that normally associates the odor with the sugar is inhibited by CFM. According to this scheme, pre-feeding with NAS can be considered as an associative gustatory conditioning, in which the lack of energy is a negative internal state that is associated to the sugar taste. CFM is encoded in the memory center, the MBs, while the peripheral taste sensitivity to sugar is not modified after NAS presentation, as shown by the normal responses in PER experiment (Fig. 1b) and sugar preference test (Supplementary Fig. 1c). CFM is thus a central brain process that can be distinguished from desensitization.

D-glucose, a classical nutritious sugar, does not elicit an over-valuation of glucose taste when presented during pre-feeding. Indeed, this would have been manifested as an increased score in comparison to the score of non-pre-fed flies, which we did not observe (Fig. 1a). Furthermore, complementing L-glucose pre-feeding with the tasteless and energetic D-sorbitol prevented devaluation without eliciting any over-valuation (Supplementary Fig. 1g). Thus, the sweet taste in itself predicts the delivery of an energy income for the *Drosophila* brain under physiological conditions. However, the situation differs from expectation in the case of NAS feeding, in that it is learned in order to adapt the behavior so that the animal is not attracted by cues associated to this taste (e.g., odor in our study). This devaluation cannot be generalized among NAS with different tastes, suggesting that flies discriminate between various sugars based on taste sensation. Nevertheless, this controversial hypothesis has not yet been clearly demonstrated at the behavioral level[29]. One recent study demonstrated that adding excess NAS to *Drosophila* food leads to abnormal hunger behavior and elicits a higher motivation to consume food[51], and the effect was attributed to an imbalanced ratio between a high sweet taste and a low energy food value. We can hypothesize that this greater hunger state could be the result of sweet food devaluation, as such flies consume more food to fill their expected energy requirement. Since CFM formation requires only one minute of NAS feeding, a prolonged treatment could lead first to CFM formation, and then to a metabolic disorder through the insulin/NPF pathway[51].

We have shown at the neuronal circuit level that CFM formation relies on the MBs as well as DPM and PAM synaptic activity during and immediately after pre-feeding. Our results suggest the involvement of an MB-DPM connectivity that is required for CFM formation, as it has been proposed for olfactory memories in *Drosophila*[37,39,41,42]. The observation that CFM is sensitive to cold-shock anesthesia is compatible with the idea that this memory depends on recurrent electrical activity. We previously showed that after sugar/odor association, the MBs "wait" for energy signaling for several hours[13]. We propose here that during NAS consumption, PAM neurons first signal the food reward signal to the MBs[33], but after consuming the NAS, the MB-DPM interaction encodes the NAS devaluation within several hours after pre-feeding. In contrast, introducing the energy signal during the consolidation period should prevent this devaluation by severing the MB-DPM communication. Recently, it was demonstrated that the MB Kenyon cells respond to sweet stimulation at the dendritic level[43]. Similarly, we demonstrate here that the DPM neurons respond to sweet stimulation. Thus, both the MBs and DPM activity are required during pre-feeding for CFM formation and display a calcium response to sweet stimulation.

Our results also demonstrate plasticity in dopaminergic MB-afferent PAM neurons. In line with behavioral results, pre-feeding flies with L-glucose induced a decrease in the PAM neuron response to L-glucose. This devaluation affects the central brain response of dopaminergic neurons to NAS. Thus, CFM mechanisms could be related to the decreased dopaminergic response observed in the mammalian midbrain when a cue is not followed by a reward, in a phenomenon referred to as prediction error signaling[52,53]. The decreased PAM response after L-glucose pre-feeding explains why flies are subsequently unable to form positive associations between olfactory stimuli and this particular sugar. The DPM also demonstrate plasticity through an increased neuron response to L-glucose following L-glucose pre-feeding. Recent experiments indicate that the DPM not only connect the MBs, but they also display a connectivity to PAM neurons[54]. Therefore, modified activity of PAM neurons during CFM retrieval (olfactory conditioning) could be a consequence of DPM

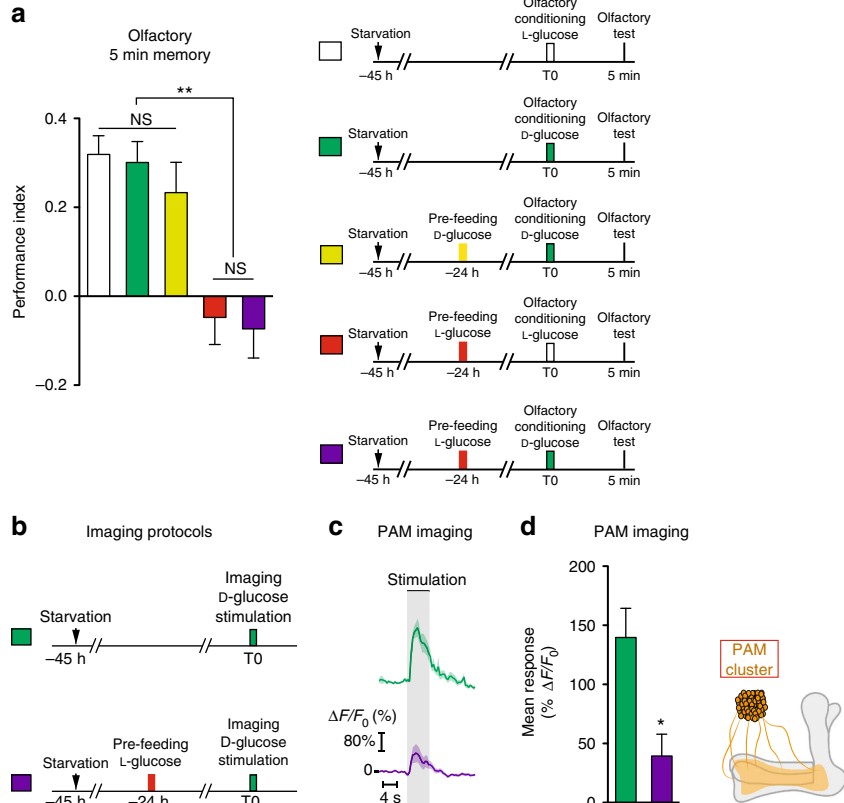

**Fig. 5** NAS pre-feeding devalues caloric sugar. **a** Pre-feeding flies for 1 min with non-nutritious L-glucose 24 h before olfactory conditioning with the same sugar or the energetic D-glucose induces significantly lower STM scores as compared to controls ($F_{(4,55)} = 12.63$; $p < 0.0001$; $n \geq 11$; $p = 0.999$ in post hoc comparison between flies not pre-fed and conditioned with L-glucose and flies not pre-fed and conditioned with D-glucose, $p = 0.920$ in post hoc comparison between flies not pre-fed and conditioned with L-glucose and flies pre-fed and conditioned with D-glucose, $p = 0.829$ in post hoc comparison between flies not pre-fed and conditioned with D-glucose and flies pre-fed and conditioned with D-glucose, $p = 0.997$ in post hoc comparison between flies pre-fed and conditioned with L-glucose and flies pre-fed with L-glucose and conditioned with D-glucose). **b** Pre-feeding protocol used before the imaging experiment: flies were pre-fed or not with L-glucose for 1 min, 24 h before D-glucose stimulation under the microscope. **c** Time course response of PAM neurons. $n = 10$. Black bar: stimulus presentation. **d** Average response to L-glucose. Flies pre-fed with L-glucose displayed a significantly lower response to caloric D-glucose in comparison to non-pre-fed flies ($t$-test, $t_{18} = 3.264$; $p = 0.004$; $n = 10$). Means are ± SEM; statistical tests: $t$-test and one-way ANOVA; NS: $p \geq 0.05$; *$p < 0.05$; **$p < 0.01$ in comparison between two groups for $t$-test and in post hoc comparisons with both parental controls for ANOVA

modulation through the release of serotonin and/or GABA[38]. This observation highlights how a high level of taste integration can be modulated by experience without affecting sugar preference at the sensory level.

Importantly, we deciphered the impact of NAS on food value in this study. Our results demonstrate that only 1 minute of exposure to L-glucose is enough to induce a long-term devaluation of this NAS. Furthermore, recent research has shown that long-term NAS consumption leads to abnormally high feeding behavior[51]. Taken together with our present study, this result highlights the significant impact of NAS on *Drosophila* feeding behavior. Ultimately, the current findings should underscore the importance of CFM formation and the impact of NAS consumption on feeding behavior in other species, especially humans.

## Methods

**Fly strains.** Fly stocks were raised on standard food at 18 °C and 60% relative humidity under a 12:12 h light:dark cycle. The Canton-Special (CS) strain was used as the wild-type *Drosophila melanogaster* strain. All transgenes were used in a CS background. Specific expression was driven in DPM neurons using *VT62446-GAL4*; in MB neurons using *VT30559-GAL4*[35]; and in subsets of MB neurons using *VT049483-GAL4* (γ neurons); *MB008B-GAL4* (α/β neurons) and *VT030604-GAL4* (α'/β' neurons). The *58E02-GAL4* and *UAS-shi^{ts}* lines were described in a previous study[13]. For the dCREB experiment we used dCREB^{RNAi}. GAL4 lines from the Flylight and VT collections were obtained from the Bloomington *Drosophila* Stock Center and the Vienna *Drosophila* Resource Center, respectively. The expression

pattern of the lines from the Flylight and Vienna Tile collections are available from the Flylight project and Brainbase websites, respectively. We used tub-GAL80^{ts} for adult transgene expression[36]. For imaging experiments, flies were raised on standard food at 25 °C and 60% relative humidity under a 12:12 h light:dark cycle. For calcium reporting, we used the *20XUAS-IVS-GCaMP3* line (in the VK00005 insertion site) and the *20XUAS-IVS-GCaMP6f* line (in the VK00005 insertion site) from the Janelia Farm Research Center[48], in addition to *UAS-GCaMP3*[49].

**Pre-feeding and training protocol.** After hatching, adult male and female flies were kept overnight in fresh bottles containing standard medium, and were then transferred at 25 °C into starvation bottles containing a cotton wool disk soaked with 6.8 ml of mineral water for 16–21 h.

For pre-feeding experiments, flies were pre-fed with either standard food medium, non-nutritious L-glucose, non-nutritious arabinose, nutritious D-glucose, a mixture of D-glucose and phlorizin, or a mixture of L-glucose and D-sorbitol. L-glucose and D-glucose were used at a 2 M concentration in mineral water; arabinose was used at a 3 M concentration in mineral water. Phlorizin was pre-dissolved in mineral water and used at a 200 mM final concentration. D-sorbitol was used at 0.01, 0.03, 0.1 and 0.3 M.

When pre-fed on standard medium, flies were transferred into normal food bottles for 30 min, and then returned to starvation bottles. For sugar pre-feeding experiments, flies were placed for 1 min in a test tube 24 h before conditioning. The tube walls were covered with either L-glucose, arabinose, D-glucose or a mixture of D-glucose and phlorizin. Flies were then returned to starvation bottles. For imaging, flies were pre-fed with either L-glucose, D-glucose, a mixture of D-glucose and phlorizin, or arabinose and imaged 24 min after training (see Fig. 4a–f; Supplementary Fig. 5a–f).

For pre-feeding followed by re-feeding (Supplementary Fig. 1f; Supplementary Fig. 5g–i), flies were pre-fed on 2M L-glucose for 1 min as usual and immediately transferred either into a starvation bottle, a D-glucose feeding tube (for 1 min), or

onto standard medium (for 30 min). The latter two groups were then transferred back into starvation bottles.

The conditioning apparatus and protocol have been previously described[19]. Briefly, groups of 30–40 flies of a given genotype were conditioned in a barrel by exposure to one odor paired with a sugar reward; subsequent exposure to a second odor took place in the absence of sugar. The sequence of a single training session consisted of an initial 90-s period of non-odorized airflow, 60 s of the first odor, 45 s of non-odorized airflow, 60 s of the second odor, and 45 s of non-odorized airflow. Odors were produced using 3-octanol ( > 95% purity; Fluka 74878, Sigma-Aldrich) at 0.360 mM, and 4-methylcyclohexanol (99% purity; Fluka 66360) at 0.325 mM diluted in paraffin oil. For the D-glucose, L-glucose, and arabinose experiments, the sugar reward was either L-glucose, D-glucose, or arabinose, with respect to a previously published protocol[19]. All sugars and phlorizin were obtained from Sigma-Aldrich. Note that a similar level of STM was formed after pairing an odorant with either nutritious or non-nutritious sugar[11].

Cycloheximide (CXM) was used to examine the formation of protein synthesis-dependent LTM[18,19]. The vehicle solution for drug feeding was mineral water (Evian). The protocol for CXM treatment followed a previously published protocol[19]. After 1 d on fresh medium, flies were transferred into 15-ml Falcon tubes containing one Whatman filter paper ($1 \times 2.5$ cm$^2$) soaked with 125 μl of 35 mM CXM solution (94% purity; Sigma, C7698) in vehicle or with vehicle alone (control) for 15–18 h at 25 °C. Drug feeding only occurred before pre-feeding. After pre-feeding, flies were kept in a regular starvation bottle for 24 h.

Cold shock treatment was used to examine whether CFM is consolidated or not. 1 h prior to olfactory conditioning, flies were transferred into empty vials and placed in an ice bath at 4 °C for 1 min. Flies were then transferred back into starvation vials at 18 °C until the beginning of the experiment.

**Memory performance test**. During the memory performance test, flies were exposed to both odors simultaneously in a T-maze for 1 min. The performance index (PI) was calculated as the number of flies attracted to the conditioned odor minus the number of flies attracted to the unconditioned odor, divided by the total number of flies in the experiment. A single PI value represents the average of the scores from two groups of genotypically identical flies trained with either octanol or methylcyclohexanol as the CS+ (i.e., an odor paired with the sugar presentation). STM was evaluated 2 h after conditioning. All memory tests were performed at 25 ° C.

**Temperature-shift protocols**. To block synaptic transmission during and after pre-feeding, flies expressing Shi$^{ts}$ were placed at the restrictive temperature (33 °C) 30 min before pre-feeding, and then moved to an incubator at the permissive temperature (18 °C) 2–3 h after pre-feeding. Permissive temperature control experiments were performed at 25 °C. Time courses of the temperature shifts employed in each experiment are provided alongside each relevant graph of memory performance. For memory experiments, flies were stored at 18 °C after pre-feeding, prior to training. For RNAi expression in adult MBs, flies were maintained at 30 °C for 2 days prior to LTM training at 25 °C. For non-induced LTM experiments, flies were placed at 18 °C for 2 days prior to training at 25 °C. Flies were stored at 18 °C after acquisition, prior to testing at 25 °C. For pre-feeding experiments, flies were maintained at 30 °C for 2 days prior to pre-feeding for induction, or at 18 °C for controls. After the pre-feeding step, experiments were performed as in the classic protocol.

**Proboscis extension reflex test**. For tarsal proboscis extension reflex (PER), flies were mounted on glass slides using nail polish. Flies were allowed 1–2 h to recover before testing began. Flies were stimulated with water on their front tarsi and allowed to drink until satiated. Each fly was then stimulated with a tastant on the tarsi, and responses to each trial was recorded. Flies were provided with water between each tastant. All stimuli were delivered with a 1-ml syringe attached to a 20-μl pipette tip. Fly responses were tested on L-glucose at increasing concentrations: 1, 10, 100, 500 mM and 2 M. Each L-glucose concentration was only delivered once to each fly.

**Sugar response tests**. Tests were performed with a T-maze apparatus as previously described[24]. One maze arm with L-glucose was tested against one empty maze arm, in an odorless airflow. Flies were trapped in either arm after 1 min in the dark. The L-glucose arm was placed alternately on the right or left. The L-glucose response was calculated as for the memory test and then used as a score. We prepared the L-glucose arm as follows: an L-glucose solution in mineral water (Evian) was applied on a band in the inner surface of plastic test tubes, using a piece of imitation felt (0.5–0.6 cm) soaked with 0.4 ml L-glucose solution (three tubes prepared with one felt). If not noted otherwise, the L-glucose concentration was 0.2 M. Tubes were left at room temperature for 18–28 h before testing to allow the sugar to dry. Each tube was used for four consecutive tests. The sugar response tests were performed at the restrictive temperature for flies carrying the UAS-shi$^{ts}$ transgene (33 °C), and at 25 °C for CS flies.

**Olfactory acuity**. Tests were performed as previously described[19] at 25 °C for CS flies. Flies were starved for 21 h and then pre-fed 24 h before the olfactory test. Each

odor was tested for 1 min against its solvent (paraffin oil). The response index was calculated as for the memory response test and then used as a score. The odor was delivered alternately through the right or left arm of the maze. A PI of 1 indicates complete behavioral repulsion.

**In vivo calcium imaging**. In vivo confocal imaging and subsequent data analysis of sugar stimulation were performed following previously described protocols[12,33]. For PAM imaging, images were acquired at a rate of one image every 500 ms. Female flies of the genotype $w^{1118}/w^{1118}$; UAS-GCaMP3/+; 58E02-GAL4 were used to image PAM neurons. For stimulation, an 8-μl droplet of mineral water (2 M L-glucose or 2 M D-glucose solution in mineral water) was deposited on a plastic plate and brought within reaching distance of the fly for 5 s via a micromanipulator. For DPM imaging, GCaMP6f fluorescence was viewed with a Leica SP5 II laser scanning confocal microscope equipped with a tandem scanner and HyD detector. The relevant area of the DPM was visualized using the 25 × water objective with an electronic zoom of 2.5. Images were acquired at a speed of 8000 lines per second with a line average of four, and at a rate of one image every 200 ms. Female flies of the genotype $w^{1118}/w^{1118}$, UAS-IVS-GCaMP6f/+; VT64246-GAL4/+ were used to image DPM neurons. For stimulation, an 8-μl droplet of mineral water (2 M L-glucose or 2 M D-glucose solution) in mineral water was deposited on a plastic plate and brought within reaching distance of the fly for 5 s via a micromanipulator.

Image analysis was performed essentially as previously described[12,33]. For each region of interest, the baseline ($F_0$) was estimated as the mean fluorescence over the 10 frames preceding the stimulus, and the mean response was calculated as the average of $\Delta F/F_0$ during the period when the droplet of solution was available. Responses from both hemispheres were averaged to yield the mean response of each fly; for time courses, both hemispheres were considered for each animal. At least eight flies were tested per condition and then averaged.

**Data analysis and statistics**. All data are presented as means ± SEM. A two-tailed unpaired $t$-test was used to compare the data series between two conditions. Results of the $t$-test are given as the value $t_x$ for the $t$ distribution with $x$ degrees of freedom obtained from the data. Comparisons between more than two distinct groups were analyzed by one-way ANOVA. ANOVA results are given as the value of the Fisher distribution $F_{(x,y)}$ obtained from the data, where $x$ is the number of degrees of freedom between groups (one-way ANOVA) and y is the residual number of degrees of freedom. ANOVA was followed by pairwise planned comparisons between relevant groups with a Student's–Newman–Keuls test. Asterisks denote the smallest significant difference between the relevant group and its genotypic controls, using post hoc pairwise comparisons (*$p < 0.05$, **$p < 0.01$, and ***$p < 0.001$; NS, not significant). A repeated-measure analysis of variance (RM-ANOVA) was used to analyze responses to increased concentrations of L-glucose (PER), with concentrations (from 1 mM to 1 M) and pre-feeding conditions (no pre-feeding/L-glucose pre-feeding/D-glucose pre-feeding/medium pre-feeding/D-glucose+ phlorizin pre-feeding) as the within-group factors. Monte Carlo simulations demonstrated that it is permissible to use ANOVA on dichotomous data under controlled conditions[55].

**Data availability**. The data that support the findings of this study are available from the corresponding author upon reasonable request.

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

## Acknowledgements

T.P. received grant support from the French Agence Nationale de la Recherche (11BSV40071 and ANR-15-CE32-0008-01). We thank all the members of our laboratory for their valuable discussions and for critically reading this manuscript.

## Author contributions

P.-Y.M. and A.L.-S.-A. performed the experiments. P.T. and T.P. supervised the work. P.-Y.M., P.T. and T.P. designed the experiments and interpreted the results. P.-Y.M. and T.P. wrote the manuscript.

## Additional information

**Competing interests:** The authors declare no competing financial interests.

