## [Peer Review File · Nature Communications]

Reviewers' expertise:

Reviewer #1: Sugar and nutrient sensing in *Drosophila*;

Reviewer #2: Sugar and nutrient sensing in *Drosophila*;

Reviewer #3: *Drosophila* learning and memory, mushroom bodies.

Reviewers' comments:

Reviewer #1 (Remarks to the Author):

Musso and colleagues introduce a new assay to reveal appetitive reward memory formation in the fruitfly *Drosophila*. The paradigm relies on pre-exposure of flies to non caloric artificial sweetener (NAS) or a nutritious sugar, followed by conditioning 24 hrs later in which the NAS is associated with an odor, and finally testing memory formation another 2 hrs later using a standard T-maze. The authors show that flies generate a robust memory when pre-exposed to a nutritious sugar, but fail to do so when pre-exposed to NAS, a memory they refer to as caloric frustration memory (CFM). They suggest that this memory is different from well-established long term memory which is also a sugar-mediated appetitive reward memory. They claim that CFM is insensitive to cycloheximide mediated suppression of protein synthesis (whereas LTM requires protein synthesis). The authors show that CFM requires the mushroom bodies, and that the DPM (dorsal paired medial) neurons, previously shown to be important for forming appetitive LTM by nutritious sugars, are also necessary for CFM. Lastly, they show that Ca²⁺ responses in a subset of these neurons, referred to as PAM neurons, also previously reported to be important for appetitive LTM formation, is suppressed in flies pre-exposed to a NAS.

While the work presented here is well done overall (with the exception of one crucial experiment; see below), no convincing evidence is presented that the CFM is different from the well-established appetitive LTM that relies on nutritious sugars and is not formed by NAS. The use of a novel assay does not necessarily reveal a new type of memory, as a memory can be revealed by a number of different paradigms. CFM shares many aspects with appetitive LTM, including the involvement of PAMs. The main claim that they use to distinguish CFM from LTM is the suggestion that CFM does not rely on protein synthesis. However, the experiment for this claim is poorly executed. There is no evidence that protein synthesis inhibition by cycloheximide has occurred (Figure 2). Why was cycloheximide not presented in the 24 hrs window prior to conditioning? It is not clear how long cycloheximide remains in fly at concentration high enough to efficiently suppress protein synthesis. Recovery of protein synthesis after drug removal is rapid and occurs within 1 to 2 hours in mammalian cells (Helinek et al, 1981). Given the lack of information when (or even whether) the flies have ingested the drug makes this negative result impossible to interpret. The control experiment they present is set up differently (the drug is presented for the 24 hrs, immediately before conditioning), making it more likely that during the critical time it was present at sufficiently high concentration to be effective. Cycloheximide tastes bitter and hence, there is a natural aversion to ingest it, even when flies are starved. In order to claim that CFM is not dependent on protein synthesis, the authors need to come up with convincing evidence for activity of the drug in the delivery method applied in the relevant assay. This lack of evidence for a protein synthesis independent process and the many

similarities to LTM makes it therefore possible, or even likely, that CFM proposed here and LTM are the same, revealed through different assays.

Additional points: The introduction about the potential health benefits (or lack thereof) is somewhat irrelevant, especially at the length it is presented, since the paper does not deal with this issue at all.

To inactivate the DPM neurons, authors use a single GAL4 line expressed in many (or all) MB neurons. There are many more restricted GAL4 lines available, and it would be interesting to test whether inactivation of subpopulations of Kenyon cells is sufficient to affect CFM.

To test whether PAM neurons are involved in CFM, the authors should also use flies pre-fed with D-Glucose, not only L-Glucose.

Overall, the study does not show convincingly that the memory effect observed in the paradigm is different from the LTM that flies associate with nutritious sugars, and hence do not support the central claim of a new memory quality.

Minor points:

Abstract: any sweet taste predicts an energy income, not just NAS.

Page 3: authors refer to PER and a few lines later to approach behavior. What is the latter exactly? Cite a relevant paper.

Page 4, third and fourth to last lines in Intro: L-glucose is used twice and redundant in that sentence.

Page 6, second paragraph (lines 5 to 8); References are missing: Weiss et al., Yavuz et al., for differentiated responses of taste neurons to different sugars, and Masek and Scott, for lack of clear evidence for such.

Page 6, third paragraph (line 3 from bottom): what do they refer to with "both groups"?

Page 8/Figure 3: What does STM mean (Figure 3): Authors examine effects of inactivation of DPM neurons on CFM but Figure uses term STM (= short term memory)? Please clarify.

Page 8/9: In the sentence bridging these pages, the authors do measure the reponse of neurons, not of flies; please correct.

Page 9, paragraph 2, second line: insert the sugar used: Following the pre-feeding step with L-glucose,

Reviewer #2 (Remarks to the Author):

In the manuscript, Musso et al showed that *Drosophila* are able to learn the non-nutritional property of non-caloric artificial sweeteners (NAS) by using a combination of behavior and calcium imaging experiments. They fed flies with non-caloric L-glucose for 1 minute and trained these flies 24 hours later with an odor paired with L-glucose, and then tested them for olfactory short-term memory (STM) 2 hours later. The results were intriguing. These flies prefed with L-glucose for 1 minute showed significantly lower scores compared to flies that were not prefed with L-glucose. The authors interpreted that flies devalue NAS when they experience that palatable taste experience is not followed by nutritious intake, called

Caloric Frustration Memory (CFM). The work is novel and highly significant as the authors were able to demonstrate another form of memory using the powerful *Drosophila* system. I would recommend Nature Communications to publish this work, but have some concerns.

Concerns:

1. It would be important to demonstrate CFM using another behavior paradigm to validate this form of memory. The authors might be interested, for example, in training flies with an odor paired with L-glucose and another odor without sugar, and testing these flies for STM after 1-2 hours and LTM after 24 hours. The evidence produced thus far would predict that these flies be repelled by the odor associated with L-glucose when tested for LTM, whereas the flies be attracted to the odor when tested for STM.
2. The authors' previous work showed that flies are capable of forming LTM after conditioning with L-glucose if D-glucose or nutritive sugar is provided within 5 hours after being exposure to L-glucose. As such, a brief exposure to D-glucose within 5 hours should abolish CFM. They can address this by using behavior and calcium imaging experiments.
3. The authors claims that devaluation is not a sensory process because flies pre-fed with L-glucose display normal responses to L-glucose, illustrated in Figure 1B. This argument is feeble because it may not be conceivable that L-glucose tastes same as D-glucose+Phlorizin to flies. The authors need to use mutant flies that are insensitive to sweet taste or/and use a more sensitive behavior assay such as PER response assay to address this.
4. The authors can mix different amounts of non-sweet, but caloric sugar such as sorbitol with L-glucose and pre-feed flies to determine a dosage-response curve.
5. The authors need to better describe the T-maze experiments shown in Figure 1B, Supplementary Figures 3B, 3D and 1A.
6. It may be better to use PER assay instead of the T-maze experiments to determine whether or not behavioral responses of tested flies to L-glucose or olfactory cues are normal.
7. I would suspect that pre-feeding flies with L-glucose or D-glucose+Phlorizin would affect the preference to L-glucose in Figure 1B, as was the case in the conditioning experiments in Figure 1A. Do the authors have an explanation for this?
8. In Figure 4, the authors should examine the flies prefed with D-glucose, regular fly medium and D-glucose+Phlorizin in calcium imaging to support that the correlation expands to other sugars.
9. The gender of tested flies was not described. Prolonged periods of starvation have vastly different effects on males versus females.
10. Why were flies tested 2 hours after training for STM in Figure 1A, whereas flies were tested 5 minutes after training in Supplementary Figure 1B? The PI was lower than zero when tested immediately after training.

Reviewer #3 (Remarks to the Author):

Musso et al. characterize a novel form of memory, which they call Calorie Frustration Memory (CFM). CFM forms after flies are fed non-caloric artificial sweeteners (NAS), and consists of a sustained reduction in the ability of flies to form appetitive associations with

sweet, but non-nutritious, rewards. CFM is an anesthesia sensitive memory that lasts over 24 hrs, does not seem to require new protein synthesis, and requires activity of mushroom body Kenyon cells, and DPM cells. The authors further show that feeding flies a NAS, L-glucose, leads to a subsequent reduction in L-glucose-dependent calcium influx in PAM dopaminergic neurons. Since PAM neurons are required for formation of appetitive associations, the authors propose that this decrease is responsible for CFM. This paper is interesting and provides some insights into the roles of DPM, dopaminergic, and mushroom body neurons and their connections in CFM formation. However, there are several points that are unclear that I would like the authors to address experimentally.

1) The experiments in Figure 1B are critical for understanding CFM and should be expanded. In particular, the specificity of the associations affected by CFM, and the effect/non-effect of CFM on L-glucose preference should be better characterized.

A) Figure 1B is difficult to understand. The authors suggest that, 1) flies can distinguish between different sugars based on taste, and 2) flies can determine whether a particular sugar is nutritious or not. Consistent with this, in Figure 1A, the authors show that formation of appetitive associations between an odor and a taste (L-glucose) is inhibited. This inhibition likely occurs because L-glucose causes calorie frustration. However, in Figure 1B, the authors show that calorie frustration does not affect attraction to L-glucose itself. Thus, flies are somehow prevented from forming appetitive associations with a non-nutritious sugar without affecting their attraction to the sugar. This is counterintuitive. Why is the association between an odor and calorie frustration inhibited, while the more direct association between a taste and calorie frustration unaffected? If the authors propose that the attraction to L-glucose is innate, they should demonstrate that the results in Figure 1B occur independently of mushroom body activity.

B) In Figure 1B, the authors measure L-glucose preference by making flies choose between L-glucose and water, both of which are non-nutritious. This is a choice between two non-beneficial situations, so this choice parameter may not be sensitive enough to detect changes in L-glucose preference. If the same experiment is repeated with the choice between L-glucose and nutritious medium, is the L-glucose preference altered by pre-feeding conditions?

C) Does pre-feeding with D-glucose in the absence of Phlorizin increase L-glucose preference?

2) Results in Figure 1A suggest that pre-feeding with L-glucose should impair later olfactory conditioning with D-glucose, at least if tested at 5 min. This seems contrary to optimal behavior, but should occur if the authors model is correct. Does this happen?

3) Authors suggest that activity of DPM neurons is required for formation of CFM and appetitive memory. They further show that L-glucose and D-glucose feeding induces calcium influx in these cells. Does calcium influx decrease after pre-feeding L-glucose?

4) The role of PAM-dopaminergic neurons in CFM is not clear. Do the authors think that PAM activity is important during CFM formation? If so, does blocking PAM output during pre-feeding of L-glucose inhibit CFM formation? Also, if the authors believe that reduced PAM activity during training is responsible for CFM, does artificial activation of PAM-DANs abolish

CFM?

Minor points.

Please describe the response of DPM neurons to L/D-glucose more precisely. Since DPM neurons respond to rewards as well as punishments, are calcium responses to different stimuli confined to distinct compartments on the MB lobes?

Please explain why 5 min memory to L-glucose in non-pre-fed flies is lower than 2 hr memory (Fig. 1A).

The authors conclude that DPM neurons are involved in CFM, which is an anesthesia-sensitive form of memory. However, Lee et al. (PNAS 2011) show that DPM neurons are required for anesthesia-resistant memory, but not anesthesia-sensitive memory. How do the authors explain this discrepancy?

We thoroughly revised our manuscript to address the points that have been raised by the reviewers, adding many additional experimental data, as detailed below. We hope that this revised version now adequately responds to each reviewer's concerns.

Reviewer #1 (Remarks to the Author):

Musso and colleagues introduce a new assay to reveal appetitive reward memory formation in the fruitfly *Drosophila*. The paradigm relies on pre-exposure of flies to non caloric artificial sweetener (NAS) or a nutritious sugar, followed by conditioning 24 hrs later in which the NAS is associated with an odor, and finally testing memory formation another 2 hrs later using a standard T-maze. The authors show that flies generate a robust memory when pre-exposed to a nutritious sugar, but fail to do so when pre-exposed to NAS, a memory they refer to as caloric frustration memory (CFM). They suggest that this memory is different from well-established long term memory which is also a sugar-mediated appetitive reward memory. They claim that CFM is insensitive to cycloheximide mediated suppression of protein synthesis (whereas LTM requires protein synthesis). The authors show that CFM requires the mushroom bodies, and that the DPM (dorsal paired medial) neurons, previously shown to be important for forming appetitive LTM by nutritious sugars, are also necessary for CFM. Lastly, they show that Ca²⁺ responses in a subset of these neurons, referred to as PAM neurons, also previously reported to be important for appetitive LTM formation, is suppressed in flies pre-exposed to a NAS.

While the work presented here is well done overall (with the exception of one crucial experiment; see below), no convincing evidence is presented that the CFM is different from the well-established appetitive LTM that relies on nutritious sugars and is not formed by NAS. The use of a novel assay does not necessarily reveal a new type of memory, as a memory can be revealed by a number of different paradigms. CFM shares many aspects with appetitive LTM, including the involvement of PAMs. The main claim that they use to distinguish CFM from LTM is the suggestion that CFM does not rely on protein synthesis. However, the experiment for this claim is poorly executed. There is no evidence that protein synthesis inhibition by cycloheximide has occurred (Figure 2). Why was cycloheximide not presented in the 24 hrs window prior to conditioning? It is not clear how long cycloheximide remains in fly at concentration high enough to efficiently suppress protein synthesis. Recovery of protein synthesis after drug removal is rapid and occurs within 1 to 2 hours in mammalian cells (Helinek et al, 1981). Given the lack of information when (or even whether) the flies have ingested the drug makes this negative result impossible to interpret. The control experiment they present is set up differently (the drug is presented for the 24 hrs, immediately before conditioning), making it more likely that during the critical time it was present at sufficiently high concentration to be effective. Cycloheximide tastes bitter and hence, there is a natural aversion to ingest it, even when flies are starved. In order to claim that CFM is not dependent on protein synthesis, the authors need to come up with convincing evidence for activity of the drug in the delivery method applied in the relevant assay. This lack of evidence for a protein synthesis independent process and the many similarities to LTM makes it therefore possible, or even likely, that CFM proposed here and LTM are the same, revealed through different assays.

The reviewer raises an important point, which we have addressed with further explanations and additional experiments. First, we would like to emphasize that CXM is ingested by starved flies despite its bitter taste, as shown by the control experiment (Supplementary Fig. 2). Second, because

of the time required to ingest CXM, and the time required for CXM to affect protein synthesis-dependent processes, the drug is classically delivered to flies within the 24-hr window that precedes training (Krashes and Waddell 2008; Colomb et al., 2009). Thus, in our case CXM was given in the 24 hrs prior to non-energetic sugar pre-feeding, since CFM is formed following this pre-feeding. The olfactory conditioning mentioned by referee #1 serves to reveal CFM, but olfactory memory is not the target of CXM. Therefore, CXM was not given in the interval between pre-feeding and olfactory conditioning. Third, to further demonstrate that CFM differs from LTM we added an experiment in which we inhibited the expression of the transcription factor dCREB in adult MBs. Our results indicate that this did not affect CFM, while appetitive LTM was impaired (see Supplementary Fig. 4a-c). Lastly, we wish to stress that CFM is sensitive to cold-shock, which is not the case for appetitive LTM (Krashes and Waddell 2008). Therefore, we believe that our data strongly support the view that CFM does not correspond to classical appetitive LTM.

Additional points: The introduction about the potential health benefits (or lack thereof) is somewhat irrelevant, especially at the length it is presented, since the paper does not deal with this issue at all.

In accordance with the reviewer's comment, we have diminished the part dedicated to humans. However, we have chosen to retain some of this text to frame our *Drosophila* work in a more global context, as expected for the *Nature Communications* readership.

To inactivate the DPM neurons, authors use a single GAL4 line expressed in many (or all) MB neurons. There are many more restricted GAL4 lines available, and it would be interesting to test whether inactivation of subpopulations of Kenyon cells is sufficient to affect CFM.

We addressed this point by using GAL4 constructs that specifically target MB sub-populations. None of these experiments impaired CFM (Supplementary Fig. 3c-e). Thus, it appears that the global MB population is required for CFM.

To test whether PAM neurons are involved in CFM, the authors should also use flies pre-fed with D-Glucose, not only L-Glucose.

We have now investigated the PAM neural response to L-glucose after pre-feeding with D-glucose. No change of response was observed after L-glucose pre-feeding (Supplementary Fig. 5a-c).

Overall, the study does not show convincingly that the memory effect observed in the paradigm is different from the LTM that flies associate with nutritious sugars, and hence do not support the central claim of a new memory quality.

The main new message of our work is that NAS are not neutral to the brain, but instead they have long-lasting effects that affect brain plasticity. The importance of this new message remains whether CFM is similar to LTM or not. Nevertheless, we addressed the issue raised by the reviewer.

LTM is characterized by several features: (i) it depends on *de novo* protein synthesis; (ii) it requires the activity of dCREB in the MBs; and (iii) it is resistant to cold-shock anesthesia treatment (this last point was not clearly formulated in the previous version of our manuscript). On the contrary, we show here that CFM does not rely on either *de novo* protein synthesis or on dCREB activity in the

MBs, and that it is sensitive to cold shocks. We believe that our data provide good evidence that CFM does not correspond to classical LTM.

Minor points:

All the minor points from reviewer #1 have been addressed and fixed.

Abstract: any sweet taste predicts an energy income, not just NAS.

Page 3: authors refer to PER and a few lines later to approach behavior. What is the latter exactly? Cite a relevant paper.

Page 4, third and fourth to last lines in Intro: L-glucose is used twice and redundant in that sentence.

Page 6, second paragraph (lines 5 to 8); References are missing: Weiss et al., Yavuz et al., for differentiated responses of taste neurons to different sugars, and Masek and Scott, for lack of clear evidence for such.

We apologize for omitting these important contributions; we modified the corresponding paragraph and cited these works.

Page 6, third paragraph (line 3 from bottom): what do they refer to with “both groups”?

Page 8/Figure 3: What does STM mean (Figure 3): Authors examine effects of inactivation of DPM neurons on CFM but Figure uses term STM (= short term memory)? Please clarify.

As explained in the manuscript (p. 4), we used olfactory STM (Short Term Memory) to reveal the CFM.

Page 8/9: In the sentence bridging these pages, the authors do measure the response of neurons, not of flies; please correct.

Page 9, paragraph 2, second line: insert the sugar used: Following the pre-feeding step with L-glucose,

Reviewer #2 (Remarks to the Author):

In the manuscript, Musso et al showed that Drosophila are able to learn the non-nutritional property of non-caloric artificial sweeteners (NAS) by using a combination of behavior and calcium imaging experiments. They fed flies with non-caloric L-glucose for 1 minute and trained these flies 24 hours later with an odor paired with L-glucose, and then tested them for olfactory short-term memory (STM) 2 hours later. The results were intriguing. These flies prefed with L-glucose for 1 minute showed significantly lower scores compared to flies that were not prefed with L-glucose. The authors interpreted that flies devalue NAS when they experience that palatable taste experience is not followed by nutritious intake, called Caloric Frustration Memory (CFM). The work is novel and highly

significant as the authors were able to demonstrate another form of memory using the powerful *Drosophila* system. I would recommend Nature Communications to publish this work, but have some concerns.

Concerns:

1. It would be important to demonstrate CFM using another behavior paradigm to validate this form of memory. The authors might be interested, for example, in training flies with an odor paired with L-glucose and another odor without sugar, and testing these flies for STM after 1-2 hours and LTM after 24 hours. The evidence produced thus far would predict that these flies be repelled by the odor associated with L-glucose when tested for LTM, whereas the flies be attracted to the odor when tested for STM.

We have now performed the requested experiment (Supplementary Fig. 1a, b). As expected, flies tested for STM displayed normal olfactory scores. However, at 24 hr, flies conditioned on L-glucose displayed low scores, confirming that CFM is a long-lasting memory. Our experiments demonstrate that CFM consists of the devaluation of a sugar's positive value to a neutral value, but not to a negative value. In fact, a neutral value would have been expected if the NAS was perceived as any other non-energetic molecule.

2. The authors' previous work showed that flies are capable of forming LTM after conditioning with L-glucose if D-glucose or nutritive sugar is provided within 5 hours after being exposure to L-glucose. As such, a brief exposure to D-glucose within 5 hours should abolish CFM. They can address this by using behavior and calcium imaging experiments.

We thank the reviewer for this interesting proposal, and we have performed the experiment. As expected, pre-feeding L-glucose followed by D-glucose exposure rescued CFM (Supplementary Fig. 1f). Furthermore, the same protocol executed on PAM imaging rescued the PAM response to L-glucose (Supplementary Fig. 5g-i).

3. The authors claims that devaluation is not a sensory process because flies pre-fed with L-glucose display normal responses to L-glucose, illustrated in Figure 1B. This argument is feeble because it may not be conceivable that L-glucose tastes same as D-glucose+Phlorizin to flies. The authors need to use mutant flies that are insensitive to sweet taste or/and use a more sensitive behavior assay such as PER response assay to address this.

We agree on this particular point, and it is probably true that L-glucose and a mixture of D-glucose+phlorizin do not taste exactly the same. However, we respectfully contend that if two tastes are similar enough, potentially with a similar Gr activation pattern, that the CFM could be retrieved. Using mutant flies that are insensitive to taste such as *poxn* could be problematic, since taste is required for the olfactory conditioning used to reveal the CFM. To address the reviewer's concerns, we performed PER using L-glucose after the flies were pre-fed in the conditions as in Figure 1a. All groups displayed similar PER ratios to each of the L-glucose concentrations tested (Fig. 1b), strongly suggesting that CFM does not affect sugar sensitivity.

4. The authors can mix different amounts of non-sweet, but caloric sugar such as sorbitol with L-glucose and pre-feed flies to determine a dosage-response curve.

We have now performed this experiment (Supplementary Fig. 1g). As expected, adding increasing concentrations of sorbitol gradually decreased the strength of the CFM effect. This experiment demonstrates that it is indeed the lack of energetic content following a sweet experience that leads to CFM.

5. The authors need to better describe the T-maze experiments shown in Figure 1B, Supplementary Figures 3B, 3D and 1A.

The T-maze experiments are now better described (p. 19).

6. It may be better to use PER assay instead of the T-maze experiments to determine whether or not behavioral responses of tested flies to L-glucose or olfactory cues are normal.

The T-maze is classically used for control experiments in the field of appetitive learning. But we understand the point raised by the referee, and we have addressed this concern by performing PER to assess sugar responses after different pre-feeding protocols (Fig. 1b). Nonetheless, we fail to see how PER could be a better test of olfactory cues.

7. I would suspect that pre-feeding flies with L-glucose or D-glucose+Phlorizin would affect the preference to L-glucose in Figure 1B, as was the case in the conditioning experiments in Figure 1A. Do the authors have an explanation for this?

Our explanation is that CFM, by associating a sweet taste to a lack of nutrient, is a form of memory that involves the activity of the central brain circuit PAM-MB-DPM. On the contrary, peripheral activity triggers the L-glucose response, as further supported by the newly added PER experiments (Fig. 1b).

8. In Figure 4, the authors should examine the flies pre-fed with D-glucose, regular fly medium and D-glucose+Phlorizin in calcium imaging to support that the correlation expands to other sugars.

We agree that this experiment, which was also requested by reviewer #1, is important and we have now performed it (Supplementary Fig. 5). We found in flies pre-fed with D-glucose or regular fly medium that the PAM neurons displayed a high calcium response to L-glucose stimulation, but the PAM neural response to L-glucose stimulation of flies pre-fed with a mixture of D-glucose+phlorizin was significantly lower than the two previous groups. Thus, our imaging data correlate with the behavioral observations.

9. The gender of tested flies was not described. Prolonged periods of starvation have vastly different effects on males versus females.

Groups of males and females were used for all behavior experiments, as now specified (p. 16). For the calcium imaging experiment, only females were used as mentioned (p. 20).

10. Why were flies tested 2 hours after training for STM in Figure 1A, whereas flies were tested 5 minutes after training in Supplementary Figure 1B? The PI was lower than zero when tested immediately after training.

The 2 hr testing after training was used for the reason of practical convenience. Indeed, our scores at 5 min were low for an unknown reason. We performed the experiment again (Supplementary Fig. 1e) and obtained positive scores.

Reviewer #3 (Remarks to the Author):

Musso et al. characterize a novel form of memory, which they call Calorie Frustration Memory (CFM). CFM forms after flies are fed non-caloric artificial sweeteners (NAS), and consists of a sustained reduction in the ability of flies to form appetitive associations with sweet, but non-nutritious, rewards. CFM is an anesthesia sensitive memory that lasts over 24 hrs, does not seem to require new protein synthesis, and requires activity of mushroom body Kenyon cells, and DPM cells. The authors further show that feeding flies a NAS, L-glucose, leads to a subsequent reduction in L-glucose-dependent calcium influx in PAM dopaminergic neurons. Since PAM neurons are required for formation of appetitive associations, the authors propose that this decrease is responsible for CFM. This paper is interesting and provides some insights into the roles of DPM, dopaminergic, and mushroom body neurons and their connections in CFM formation. However, there are several points that are unclear that I would like the authors to address experimentally.

1) The experiments in Figure 1B are critical for understanding CFM and should be expanded. In particular, the specificity of the associations affected by CFM, and the effect/non-effect of CFM on L-glucose preference should be better characterized.

A) Figure 1B is difficult to understand. The authors suggest that, 1) flies can distinguish between different sugars based on taste, and 2) flies can determine whether a particular sugar is nutritious or not. Consistent with this, in Figure 1A, the authors show that formation of appetitive associations between an odor and a taste (L-glucose) is inhibited. This inhibition likely occurs because L-glucose causes calorie frustration. However, in Figure 1B, the authors show that calorie frustration does not affect attraction to L-glucose itself. Thus, flies are somehow prevented from forming appetitive associations with a non-nutritious sugar without affecting their attraction to the sugar. This is counterintuitive. Why is the association between an odor and calorie frustration inhibited, while the more direct association between a taste and calorie frustration unaffected? If the authors propose that the attraction to L-glucose is innate, they should demonstrate that the results in Figure 1B occur independently of mushroom body activity.

CFM formation is a central brain process that involves the MBs (Fig. 3a). As demonstrated in Supplementary Fig. 3b, inhibiting the MBs during the L-glucose preference test does not affect the preference for L-glucose.

B) In Figure 1B, the authors measure L-glucose preference by making flies choose between L-glucose and water, both of which are non-nutritious. This is a choice between two non-beneficial situations, so this choice parameter may not be sensitive enough to detect changes in L-glucose preference. If

the same experiment is repeated with the choice between L-glucose and nutritious medium, is the L-glucose preference altered by pre-feeding conditions?

In order to address this point we performed PER with increasing concentrations of L-glucose, following the pre-feeding protocols (Fig. 1b). These results confirm that the response to L-glucose is normal at the sensory level.

C) Does pre-feeding with D-glucose in the absence of Phlorizin increase L-glucose preference?

To address this question, we demonstrated in PER experiments as well as in L-glucose choice tests that pre-feeding with D-glucose does not increase L-glucose preference (Fig. 1b).

2) Results in Figure 1A suggest that pre-feeding with L-glucose should impair later olfactory conditioning with D-glucose, at least if tested at 5 min. This seems contrary to optimal behavior, but should occur if the authors model is correct. Does this happen?

We have addressed this interesting point with both behavioral experiments and PAM imaging. Strikingly, pre-feeding flies with L-glucose induced a devaluation of D-glucose when tested at 5 min. Furthermore, the PAM response to D-glucose was devaluated after L-glucose pre-feeding (Fig. 5). As mentioned by reviewer #2, this may appear to contradict optimal behavior, considering that D-glucose is energetic. But since digestion and energy signaling coming from the food source are not integrated immediately, it could be expected. Finally, testing with a longer delay might lead to a re-evaluation of the glucose taste and re-associate it with energy, finally abolishing CFM, as shown below.

3) Authors suggest that activity of DPM neurons is required for formation of CFM and appetitive memory. They further show that L-glucose and D-glucose feeding induces calcium influx in these cells. Does calcium influx decrease after pre-feeding L-glucose?

In accordance with the reviewer's comment, we now show that the response of DPM neurons to L-glucose increases after L-glucose pre-feeding (Fig. 4d). A recent publication demonstrates that DPM neurons display synaptic contact onto PAM neurons at the MB level (Takemura et al., 2017). Thus, it is possible that the inhibited response of PAM neurons due to CFM is a consequence of an inhibitory output from the GABAergic DPM (Haynes et al., 2015).

4) The role of PAM-dopaminergic neurons in CFM is not clear. Do the authors think that PAM activity is important during CFM formation? If so, does blocking PAM output during pre-feeding of L-glucose inhibit CFM formation?

It is thought that dopaminergic PAM provide a reward signal to the MBs when the fly consumes sugars. Therefore, we hypothesized that this signal is necessary to the MBs to encode the identity of the consumed sugar. Inhibiting PAM activity during pre-feeding using *Shibire^{ts}* should prevent the coding of the food identity and thereby prevent the association between the sweet taste and the lack of energy. We have performed the suggested experiment, which shows that PAM activity is indeed required during pre-feeding for CFM formation (Fig. 3e).

Also, if the authors believe that reduced PAM activity during training is responsible for CFM, does artificial activation of PAM-DANs abolish CFM?

The artificial activation of PAM during olfactory conditioning could indeed rescue CFM. In this situation, PAM activation would have to occur immediately after the sugar presentation. Currently, such an experiment is not technically feasible with our conditioning set-up and use of thermogenetics. To bypass this problem, one could consider activating the PAM before, during and after olfactory conditioning. Unfortunately, activating the PAM during the entire olfactory conditioning would temporally decorrelate the PAM activation from the taste sensation and lead to an association between the odor and the reward signaling from the PAM activity, thus creating a new memory that would mask a potential CFM rescue.

Minor points.

Please describe the response of DPM neurons to L/D-glucose more precisely. Since DPM neurons respond to rewards as well as punishments, are calcium responses to different stimuli confined to distinct compartments on the MB lobes?

The original publications that described a response of DPM neuron to electric shock did not have the spatial resolution to confine the response to specific MB compartments (Yu et al., 2005; Cervantes-Sandoval & Davis., 2012). However, the authors were able to separate the response of DPM neuron depending on their projections onto the MBs lobes. Both DPM projections onto the vertical and horizontal MBs lobes displayed an increased activity in response to electrical stimulation of the fly. When we addressed the DPM response to L-glucose and D-glucose, we originally focused on the DPM projections of the horizontal MBs lobes. In response to the reviewer's comment, we have also addressed this issue at the DPM projection of the vertical branch of the MB lobes. We thus observed that these particular projections also respond to L- and D-glucose stimulations, but do not respond when the flies are stimulated with water (see below).

Please explain why 5 min memory to L-glucose in non-pre-fed flies is lower than 2 hr memory (Fig. 1A).

To further investigate this point, we replicated the experiment (Supplementary Fig. 1e) and we observed higher 5 min memory scores equivalent to the ones obtained in Fig. 1a.

The authors conclude that DPM neurons are involved in CFM, which is an anesthesia-sensitive form of memory. However, Lee et al. (PNAS 2011) show that DPM neurons are required for anesthesia-resistant memory, but not anesthesia-sensitive memory. How do the authors explain this discrepancy?

It appears that DPM has a particularly complex role in memory processing. Indeed, DPM has been shown to be required for ARM (Lee et al., 2011), consolidation of appetitive LTM (Krashes et al., 2008), and aversive LTM (Tonoki et al., 2015), and these memories can be characterized by their resistance to anesthesia, as mentioned by the reviewer. However, DPM are also required for anesthesia-sensitive memories such as 3-hr appetitive and aversive memories (Keene et al., 2006; Pitman et al., 2013). Intriguingly, Lee et al. showed that the serotonergic activity from the DPM was specifically required for ARM, but since DPM are also GABAergic (Haynes et al., 2015) they might play a role in anesthesia-sensitive memory through the specific release of GABA.

REVIEWERS' COMMENTS:

Reviewer #1:

[**Editorial note:** this reviewer did not have any formal remarks to the authors as s/he found the revised paper to be satisfactory and recommends publication]

Reviewer #2 (Remarks to the Author):

All the suggested experiments were carried out and the results were interesting and consistent with the likely characteristics of CFM. I recommend that Nature Communications publish this work.

Reviewer #3 (Remarks to the Author):

The authors have addressed most of my concerns and the manuscript is suitable for publication. However, there is one point where I am still unclear.

The authors propose that caloric frustration memory (CFM) consists of an association between the taste of a NAS and the lack of calories, a proposal supported by their data. If this is the case, why are flies still attracted to the NAS, as demonstrated by proboscis extension and preference assays? Presumably flies do not form associations between an odor and a NAS because the NAS is no longer associated with a calorie reward. Thus it seems to me that the flies should be less attracted to the NAS. If the authors could add one or two sentences to the Discussion explaining this apparent discrepancy, I think the paper would be improved.

REVIEWERS' COMMENTS:

We thank all reviewers for their positive feedback on our revised manuscript.

Reviewer #1

{{Editor: this reviewer did not have any formal remarks to the authors as s/he found the revised paper to be satisfactory and recommends publication}}

Reviewer #2 (Remarks to the Author):

All the suggested experiments were carried out and the results were interesting and consistent with the likely characteristics of CFM. I recommend that Nature Communications publish this work.

Reviewer #3 (Remarks to the Author):

The authors have addressed most of my concerns and the manuscript is suitable for publication. However, there is one point where I am still unclear.

The authors propose that caloric frustration memory (CFM) consists of an association between the taste of a NAS and the lack of calories, a proposal supported by their data. If this is the case, why are flies still attracted to the NAS, as demonstrated by proboscis extension and preference assays? Presumably flies do not form associations between an odor and a NAS because the NAS is no longer associated with a calorie reward. Thus it seems to me that the flies should be less attracted to the NAS. If the authors could add one or two sentences to the Discussion explaining this apparent discrepancy, I think the paper would be improved.

As suggested by the reviewer, two sentences have been added to the discussion to clarify this point. As explained, CFM is a central brain process that involves mushroom bodies, while proboscis extension and sugar preference assays involve peripheral sugar sensitivity, and are not affected by NAS presentation.